# Prevalence and factors associated with puerperal sepsis among postnatal women at a Tertiary Referral Hospital in Western Uganda

Brenda Nabawanuka[1], Moses Asiimwe[1]*, Pauline Irumba[1], Julian Aryampa[1], George Wasswa[2], Michael Muhoozi[3], Peterson Amumpaire[4], Joy Acen[4], Shamim Katusabe[5], Joan Tusabe[6], Joshua Epuitai[7]

1 Mountains of the Moon University, Fort Portal, Uganda, 2 Joint Clinical Research Center, Wakiso, Uganda, 3 Baylor College of Medicine Children's Foundation, Kampala, Uganda, 4 Lira University, Lira, Uganda, 5 Makerere University Infectious Disease Institute, Kampala, Uganda, 6 Makerere University Case Western Reserve University Collaboration, Kampala, Uganda, 7 Busitema University, Jinja, Uganda

☙ These authors contributed equally to this work.
‡ PI, JA, PA, JA, SK and JT also contributed equally to this work.
* mosease19@gmail.com

## Abstract

### Background

Puerperal sepsis remains one of the leading causes of maternal mortality and morbidity in Uganda.

### Aim

This study assessed the prevalence and factors associated with puerperal sepsis among postpartum women at Fort portal Regional Referral Hospital located in western Uganda.

### Methods

A cross-sectional design was employed in the study. We conducted a records review of the patient files of 180 postnatal mothers who were admitted at Fort Portal Regional Referral Hospital from 20 February, 2024 to 01 April, 2024. A data abstraction checklist was used to collect data from participant files based on strict inclusion and exclusion criteria. Data was entered in Microsoft Excel and exported to STATA17 for data analysis. Descriptive analysis and logistic regression analysis were performed to determine the prevalence of puerperal sepsis and determinants. Bivariate and multivariate logistic regression analysis was conducted for significant factors presented as Adjusted Odds Ratios (aOR) at $p \leq 0.05$.

**Data availability statement:** All relevant data are within the manuscript and its Supporting Information files.

**Funding:** Brenda Nabawanuka obtained funding from Mountains of the Moon University Research & Innovation Fund to conduct this study. The funder had no role in study design, data collection and analysis, decision to publish, or preparation of the manuscript.

**Competing interests:** The authors have declared that no competing interests exist.

## Results

The median age of participants was 25.5 years (1QR = 20−30) and the majority (77%) had primary education. The prevalence of postnatal sepsis was 24%. Duration of hospital stay [aOR=2.30; 95%CI (1.552–3.398); $p$=<0.001], history of antepartum hemorrhage [aOR=29.09; 95% CI (1.182–716.38); $p$ = 0.039] and Anemia [aOR=0.01; 95% CI (0.001–0.218); $p$ = 0.004] were identified as factors associated with puerperal sepsis among postnatal women upon multivariate logistic regression.

## Conclusion

Puerperal sepsis was common in our setting. This study found that mode of delivery, duration of hospital stay, anemia, and Antepartum hemorrhage, were the determining factors contributing to puerperal sepsis, infection prevention measures during cesarean sections, and reducing the length of hospital stay would prove to be beneficial in the prevention of sepsis.

## Introduction

The World Health Organization (WHO) defines puerperal sepsis (PS) as an infection of the genital tract occurring at any time between the onset of rupture of membranes or labor and the 42 days after delivery in which two or more of the following are present: pelvic pain, fever, abnormal vaginal discharge, abnormal smell/ foul odor discharge or delay in uterine involution [1]. Puerperal sepsis continues to be one of the most prevalent yet preventable factors contributing to mortality [2–4]. It remains a public health concern and among the leading causes of preventable maternal mortality in developing countries, including Uganda [5]. Globally, approximately 75,000 women die each year due to puerperal sepsis [6] and by the year 2015, there were approximately 303,000 maternal death each year, with as many as 15% of these cases linked to puerperal sepsis [7]. The prevalence of puerperal sepsis varies from country to country, with higher rates observed in poorer countries, the prevalence of puerperal sepsis is 11.6% in Asia, 9.7% in Africa, 7.7% in South America, and 7.7% in Caribbean countries [7,8]. The prevalence of sepsis in Uganda varies in different health facilities. In Western Uganda, the prevalence of puerperal sepsis was 39%, [9] while a 12% prevalence was reported in Hoima District [10]. Uganda has registered a slow decline in the maternal mortality ratio (MMR) between 1990 and 2010 from 550 in 1990–438 in 2012 and from 438 to 189 per 100000 live births in 2022 [11].

The aim of the sustainable Development goal (SDG3) to lower the maternal mortality rate to 70 deaths for every 100,000 live births has not been achieved. [13]. Puerperal sepsis accounts for 10.7% of deaths among mothers and ranks as one of five leading causes maternal mortality globally after postpartum bleeding, unsafe abortion, and pregnancy-induced hypertension, puerperal sepsis is the fourth greatest cause of maternal mortality [12,13].

Factors that commonly predispose individual to puerperal sepsis include anemia, prolonged labor, repeated vaginal examinations during labor in non-sterile conditions, prolonged rupture of membranes, compromised immune function cesarean delivery and presence of retained products of conception. [2,14]. Puerperal sepsis is associated with nosocomial infections, prolonged hospitalization, increasing health expenditure, antibacterial resistance, and loss of reproductive function [15]. It has also been a major cause of anemia, puerperal psychosis, and poor lactation leading to poor infant feeding [16].

As of now, there is no published study has been done in Fort Portal Regional referral Hospital (FPRRH) or its catchment area to determine the prevalence of puerperal sepsis and its associated factors. The cases of puerperal sepsis in Fort Portal Regional Referral are on the rise in the hospital. There is no clear implementation of standardized guidelines for controlling the condition in most maternity wards at Fort Portal Regional Referral Hospital. Similarly, the factors contributing to puerperal sepsis and its preventive measures have not been studied in our setting. The available data on the occurrence of puerperal sepsis in maternity wards in Uganda is insufficient. Therefore, this study is critical in filling the knowledge gap regarding the prevalence and factors associated with puerperal sepsis among women attending the postnatal ward at Fort Portal Regional Referral Hospital. This study is also part of the baseline investigation that will later inform the implementation of a project to improve the quality and management of puerperal sepsis at Fort portal Regional Referral Hospital.

Our study determined the prevalence and factors associated with Puerperal sepsis in a regional referral Hospital in Uganda. Determining the magnitude of puerperal sepsis will help in formulating preventive measures to reduce the incidence of puerperal sepsis in the health facility. The study proposes to use the findings to develop quality improvement projects aimed at reducing the burden of puerperal sepsis in the health facility. Reducing the burden of puerperal sepsis will contribute to concerted efforts to reduce the burden of maternal morbidity and mortality in Uganda.

## Materials and methods

### Study design and area

The study was a cross-sectional analytical study. Quantitative data was collected to determine the prevalence and factors associated with puerperal sepsis. The study site was Fort Portal Regional Referral Hospital (FRRH). This is the referral hospital for the districts in Rwenzori region, including Bundibugyo, Kabarole, Kamwenge, Kasese, Ntoroko and Kyenjojo. It is a public hospital, funded by the Uganda Ministry of Health. It is one of the 13 Regional Referral Hospitals in Uganda. It has different departments including surgical ward, maternity ward, postnatal ward, medical ward, ART clinic, and gynecology department. The postnatal department admits patients with puerperal sepsis and the hospital has a 200-bed capacity.

### Study population

The study population included mothers who had delivered and had been admitted in the postnatal ward of FPRRH in a period of six months prior to data collection. These mothers were identified from the hospital database and hospital records using the inpatient identification number provided at admission. Records with duplicated records and those with incomplete data were excluded from the study.

### Sample size and sampling procedures

We used the Kish and Lisle formula for cross sectional studies, where $p = 0.1248$, was a proportion of patients who had puerperal sepsis at Hoima Regional referral hospital, Uganda [10]. A z value of 1.96 and a standard error of 0.05 was used to come up with a sample size of 180 files. All the postnatal hospital records for the previous six months prior data collection were identified and we selected all files that fulfilled the inclusion and exclusion criteria. A total of 180 files were systematically selected for the study. All data was anonymized and de-identified. Records with encrypted identities were

checked, and duplicated records or those with incomplete data were excluded from analysis. To ensure that the data is accurate and consistent, conflicting demographic information about date of birth, gender, and admission and discharge dates was investigated and resolved.

### Study variables

Independent variables considered in the analysis comprised of sociodemographic factors such as age, employment status, level of education, marital status, and address; obstetric characteristics such as parity, mode of delivery, place of delivery, duration of labor, duration of hospital stay, prolonged membrane rupture, antepartum hemorrhage/PPH retained fetal products, antenatal attendance, puerperal day of presentation episiotomy, and premature rupture of membranes.

The dependent variable in this study was puerperal sepsis. Puerperal sepsis was considered in this study as a mother with a laboratory-based diagnosis with positive bacterial culture results during the course of postnatal admission. All post-partum mothers had bacterial culture and sensitivity tests done on their samples as part of routine care with support from development partners.

### Data collection procedure

Data collection was conducted between 20 February 2024 and 01 April, 2024. We used data abstraction checklists to retrieve participant information from the hospital records. The predesigned data abstraction tool consisted of several sections including socio-demographic variables such as age, marital status, tribe, religion, level of income, level of education, obstetric characteristics such as parity, mode of delivery, number of vaginal examinations, obstetric complications such as antepartum hemorrhage, abnormal labor among others. Data comprising of patient identifiers was not retrieved for any use in this study.

### Data entry and management

Extracted data was coded, entered into Microsoft Excel, cleaned, and then imported into STATA 17 where it was analyzed. The used checklists were kept in a safe secure location under lock and key. The data in the computer was protected using a password.

### Data analysis

Descriptive statistics were used to summarize the data, frequencies, and percentages were used to summarize categorical variables, while mean and standard deviation or median and interquartile range were used to describe continuous variables. Bivariate analysis was used to determine the association between the independent variables and the presence of sepsis using a logistic regression and these were presented in form of odds ratios, confidence intervals and $p$ value. Clinically plausible variables and those with a $p$ value of ≤0.2 at bivariate analysis were taken for multivariate analyses to adjust for confounding variables. The cut off for the final factors associated with puerperal sepsis were at multivariate were those with a p value of ≤0.05.

### Quality control measures

Research assistants were trained regarding the study protocol. A pre-coded data abstraction tool was pre-tested at local public primary healthcare facility located in the study area to ensure reliability, consistency and validity appropriate changes made and thereafter used for data collection. During data collection, checklists were reviewed at the end of each participant data extraction to ensure accuracy and completeness in the data collected to ensure that corrections were made in real time. Data was cleaned and stored daily by the principal investigator.

## Ethical considerations

Ethical approval was obtained from The AIDS Support Organization (TASO) research ethics committee, TASO-2024–336. A waiver of consent was given, as data collection involved secondary data. Administrative approval from Fort portal Regional Referral Hospital and the postnatal ward in charge was sought. The study was conducted based on the ethical principles in the Declaration of Helsinki, 1964. To conduct this study, a waiver of informed consent was granted by the Independent Ethics Committee on the condition that participant data was de-identified.

## Results

### Sociodemographic characteristics of study participants

The median age of was 25.5 years with an interquartile range (IQR) of 21–30 years (Table 1). The majority of the participants were peasants or unemployed (62%), stopped in primary level of education (77%), and were in rural areas (68%). Considering obstetric characteristics, nearly a third (29%) of the participants were primigravid, more than a half (53%) were delivered by caesarean section, while a majority of them did not have any obstetric complications during pregnancy, delivery and postpartum.

**Prevalence of puerperal sepsis.** The prevalence of puerperal sepsis was 24% (43/180) (Fig 1). This underscores that nearly one-quarter of the women had puerperal sepsis.

**Factors associated with puerperal sepsis.** Logistic regression was conducted to study the factors which were significantly associated with puerperal sepsis. A bivariate logistic regression analysis was conducted (Table 2) to determine the factors associated with puerperal sepsis. In this analysis, cesarean section deliveries, longer stay in the hospital, history of premature rupture of membranes, history of antepartum hemorrhage, and belonging to other religious denominations were significantly associated with increased odds of getting puerperal sepsis.

We conducted multivariate logistic analyses to determine the factors which were significantly associated with increased odds of acquiring puerperal sepsis. Variables that were significant or were less than the probability value of 0.2 were included in the bivariate analysis. After controlling for other variables, duration in the hospital, maternal anemia, and history of antepartum hemorrhage remained significant in the multivariate analyses (Table 2). The odds of puerperal sepsis increased by more than 2 times for every unit increase in the number of days in the hospital with a 95% confidence that the true value lies between 1.5 to 3.4 times.

## Discussion

According to this study, the prevalence of sepsis at Fort Portal Regional Referral Hospital was 24% (Fig 1). This might be related to poor aseptic techniques, improper wound management practices, education provided to patients upon discharge, and an unhygienic hospital environment. Although this study shows a relatively higher prevalence than in a Uganda National Referral Hospital which reported a 12.7% [17], it is in line with the study done in Bushenyi among mothers who delivered at Kampala International University Teaching Hospital which found the prevalence to be 20.9% [18]. The findings of our study are also in line with the pooled prevalence of puerperal sepsis among postnatal women in Sub-Saharan Africa at 19.21% done by [19].

Cesarean section mode of delivery was also a significant factor associated with postpartum sepsis according to results of our study. This could be attributed to poor perioperative aseptic techniques, poor wound care practices of health care workers in post-natal units, and an unhygienic hospital environment. Poor hospital infection prevention and control practices such as poor hand hygiene practices and insufficient surgical site preparation and disinfection may potentially lead to introduction of pathogenic microbes which could potentially cause post cesarean section infection. This is in line with a study conducted in different regions from Hoima Regional Referral Hospital [20], Hawassa city, Ethiopia [21], and Tanzania [13] that revealed that having a cesarean section was associated with puerperal sepsis. However, this study

**Table 1. Socio-demographic and obstetric characteristics of the study participants.**

| Variable | Frequency n=180 | Percentage (%) |
|---|---|---|
| Age in complete years (median, IQR) | 25.5 (21-30) | |
| Marital status | | |
| Married | **120** | 66.7 |
| Single/separated/widowed | 60 | 33.3 |
| Occupation | | |
| None/Peasant | **111** | 61.7 |
| Informal/formal employment | 69 | 38.3 |
| Level of education | | |
| Primary | **139** | 77.2 |
| Secondary/tertiary | 41 | 22.8 |
| Residence | | |
| Rural | **123** | 68.3 |
| Urban | 57 | 31.7 |
| Religion | | |
| Anglican | 70 | 38.9 |
| Catholic | **88** | 48.9 |
| Others* | 22 | 12.2 |
| Parity | | |
| 1 | 53 | 29.4 |
| 2–4 | **113** | 62.8 |
| 5+ | 14 | 7.8 |
| Mode of delivery | | |
| SVD | 84 | 46.7 |
| Caesarean section | **96** | 53.3 |
| Duration of hospital stay | | |
| History of premature rupture of membranes | 4 (2-10) | |
| No | 117 | 65 |
| Yes | 63 | 35 |
| APH | | |
| No | **166** | 92.2 |
| Yes | 14 | 7.8 |
| PPH | | |
| No | **150** | 83.3 |
| Yes | 30 | 16.7 |
| Retained products of conception | | |
| No | **155** | 86.1 |
| Yes | 25 | 13.9 |
| ANC attendance | | |
| No | **95** | 52.8 |
| Yes | 85 | 47.2 |
| Episiotomy | | |
| No | **160** | 88.9 |
| Yes | 20 | 11.1 |
| Anaemia | | |
| No | **156** | 86.7 |
| Yes | 24 | 13.3 |

*(Continued)*

**Table 1.** (Continued)

| Variable | Frequency n=180 | Percentage (%) |
|---|---|---|
| HIV | | |
| No | **158** | 87.8 |
| Yes | 22 | 12.2 |
| UTI | | |
| No | **115** | 63.9 |
| Yes | 65 | 36.9 |

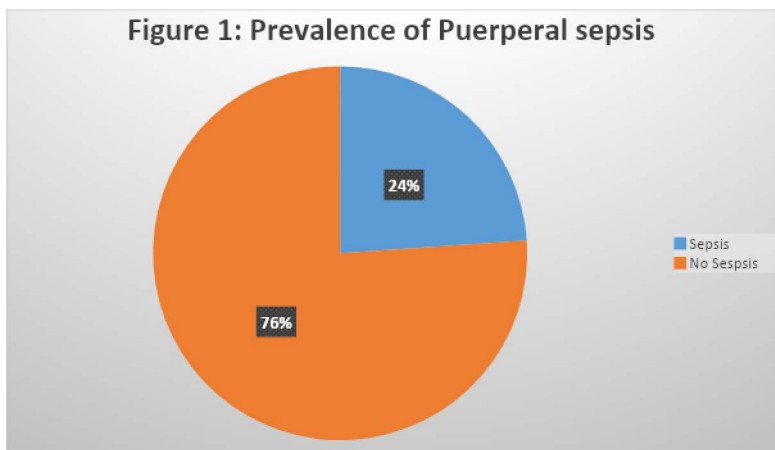

**Fig 1. A pie chart showing the prevalence of puerperal sepsis at FPRRH.**

contradicts a study that was conducted in Brazil which showed mothers who delivered by caesarean section were actually less likely to develop puerperal sepsis [22]

Findings from our study also show that anemia significantly increased the risk of puerperal sepsis. This could be attributed to poor maternal nutrition before and/or during pregnancy as well as pregnancy related infectious diseases such as malaria which is endemic in the study area. These conditions may reduce the immunity and increase susceptibility of the pregnant mother to infections. This finding is in agreement with a study conducted in Ethiopia [23]. Our results, which indicated that anemia was a significant contributing factor to puerperal sepsis, are contradicted by a study done in 2024 at the Kawempe National Referral Hospital [24].

In addition, our study shows that duration of hospital stay was a significant contributing factor to the prevalence of puerperal sepsis. The justification for the increased risk of puerperal sepsis in women who stay longer in hospital is the unhygienic hospital environment, poor aseptic processes, and hospital-acquired pathogens and infections are prevalent in the wards where the postpartum mothers are admitted during hospital stays. These results are in line with a study conducted in Ethiopia which established that prolonged hospital stay increased the risk of infection [23].

The study found that a history of antepartum hemorrhage (APH) was significantly associated with puerperal sepsis and this study is congruent with another study conducted in the same hospital [25]. Similarly, these findings are in line with the results of a Five-Year Review of Feto-maternal outcomes of Antepartum hemorrhage in a tertiary center in Nigeria which

**Table 2. Bivariate analysis of the factors associated with puerperal sepsis.**

| Variable | Sepsis (no) N = 137 | Sepsis (yes) N = 43 | OR (95% CI) | p-value = 0.05 |
|---|---|---|---|---|
| Age (median, interquartile range) | 26 (23-30) | 22 (19-30) | **0.97 (0.918-1.018)** | **0.206** |
| Marital status | | | | 0.324 |
| Married | 94 (68.6) | 26 (60.5) | Ref | |
| Single/separated/widowed | 43 (31.4) | 17 (38.5) | 1.43 (0.703-2.907) | |
| Occupation | | | | 0.213 |
| None/Peasant | 81 (59.1) | 30 (69.8) | | |
| Informal/formal | 56 (40.9) | 13 (30.2) | 0.627 (0.301-1.306) | |
| Level of education | | | | |
| Primary | 106 (77.4) | 33 (76.7) | | |
| Secondary/tertiary | 31 (22.6) | 10 (23.3) | 1.036 (0.46-2.34) | 0.932 |
| Residence | | | | |
| Rural | 95 (69.3) | 28 (65.1) | | |
| Urban | 42 (30.7) | 15 (34.9) | 1.212 (0.59-2.50) | 0.603 |
| Religion | | | | |
| Anglican | 59 (43.1) | 11 (25.6) | | |
| Catholic | 66 (48.2) | 22 (51.2) | 1.788 (0.780-3.997) | 0.157 |
| Others* | 12 (8.8) | 10 (23.3) | 4.470 (1.552-12.871) | 0.006 |
| Parity | | | | |
| 1 | 37 (27.0) | 16 (37.2) | | |
| 2–4 | 89 (65) | 24 (55.8) | 0.623 (0.298-1.307) | 0.211 |
| 5+ | 11 (8.0) | 3 (7) | 0.631 (0.155-2.570) | 0.520 |
| Mode of delivery | | | | |
| SVD | 74 (54) | 10 (23.3) | | |
| Caesarean section | 63 (46) | 33 (76.7) | 3.876 (1.771-8.483) | 0.001 |
| Duration of hospital stay in days (median, interquartile range) | 3 (1-4) | 16 (14-18) | 1.80(1.490-2.176) | <0.001 |
| History of premature rupture of membranes | | | | |
| No | 101 (73.7) | 16 (37.2) | | |
| Yes | 36 (26.3) | 27 (62.8) | 4.734 (2.290-9.785) | <0.001 |
| APH | | | | |
| No | 131 (95.6) | 35 (81.4) | | 0.005 |
| Yes | 6 (4.4) | 8 (18.6) | 4.990 (1.624-15.330) | |
| PPH | | | | 0.314 |
| No | 112 (81.8) | 38 (88.4) | | |
| Yes | 25 (18.2) | 5 (11.6) | 0.589 (0.211-1.648) | |
| Retained products of conception | | | | 0.604 |
| No | 119 (86.9) | 36 (83.7) | | |
| Yes | 18 (13.1) | 7 (16.3) | 1.285 (0.497-3.322) | |
| ANC attendance | | | | 0.420 |
| No | 70 (51.1) | 25 (58.1) | | |
| Yes | 67 (48.9) | 18 (41.9 | 0.752 (0.376-1.503) | |
| Episiotomy | | | | 0.902 |
| No | 122 (89.1) | 38 (88.4) | | |
| Yes | 15 (10.9) | 5 (11.6) | 1.070 (0.365-3.137) | |

*(Continued)*

**Table 2.** (Continued)

| Variable | Sepsis (no) N = 137 | Sepsis (yes) N = 43 | OR (95% CI) | p-value = 0.05 |
|---|---|---|---|---|
| Diabetes in pregnancy | | | | 0.904 |
| No | 130 (94.9) | 41(95.3) | | |
| Yes | 7 (5.1) | 2 (4.7) | 0.906 (0.181-4.533) | |
| Anemia | | | | 0.098 |
| No | 122 (89) | 34 (79.1) | | |
| Yes | 15 (11) | 9 (20.9) | 2.153 (0.867-5.346) | |
| HIV | | | | 0.505 |
| No | 119 (86.9) | 39 (90.7) | | |
| Yes | 18 (13.1) | 4 (9.3) | 0.678 (0.216-2.125) | |
| UTI | | | | 0.106 |
| No | 92 (67.1) | 23 (53.5) | | |
| Yes | 45 (32.9) | 20 (46.5) | 1.778 (0.885-3.570) | |
| Hypertension in pregnancy | | | | 0.926 |
| No | 128 (93.4) | 40 (93) | | |
| Yes | 9 (6.6) | 3 (7) | 1.067 (0.275-4.131) | |

found that APH was a significant predictor of puerperal sepsis among post-natal mothers [26]. APH occurs due to trauma and abnormalities of the placenta and uterus and often requires emergency modalities of delivery which may put the mother at risk of puerperal sepsis.

## Study limitations

Our study used secondary data retrieved from patients' files. Owing to the inherent nature of such data whose primary purpose was not to answer a specific research question, some patient files lacked critical data on certain variables. We did not include patient files that had missing data on any of the proposed research variables.

## Conclusion and recommendations

This investigation identified the following as risk factors for puerperal sepsis, antepartum hemorrhage, anemia, prolonged hospital stays, and mode of delivery. Thus, we advise obstetric care providers to strictly adhere to asepsis guidelines in the management of women who are likely to experience puerperal sepsis. In addition, medical professionals should manage these patients appropriately to reduce hospital stays and offer thorough prenatal education, with a focus on the significance and adherence to iron supplementation.

## Supporting information

**S1 Data. Data analyzed in this study.**
(XLSX)

## Acknowledgments

We are grateful for the support provided by Fort Portal Regional Referral Hospital's administration and staff, especially the ones working in the postnatal ward. We also want to express our gratitude to our research assistants who reviewed documents to obtain the data.

## Author contributions

**Conceptualization:** Brenda Nabawanuka, Moses Asiimwe, Shamim Katusabe, Joan Tusabe.

**Data curation:** George Wasswa, Michael Muhoozi.

**Formal analysis:** Michael Muhoozi, Joshua Epuitai.

**Funding acquisition:** Brenda Nabawanuka.

**Investigation:** Brenda Nabawanuka, Michael Muhoozi, Peterson Amumpaire, Joan Tusabe.

**Methodology:** Moses Asiimwe, Peterson Amumpaire, Joy Acen, Joshua Epuitai.

**Project administration:** Brenda Nabawanuka.

**Validation:** Moses Asiimwe, Julian Aryampa.

**Visualization:** Pauline Irumba, Julian Aryampa, George Wasswa, Joan Tusabe, Joshua Epuitai.

**Writing – original draft:** Moses Asiimwe.

**Writing – review & editing:** Moses Asiimwe, Pauline Irumba, Julian Aryampa, George Wasswa, Peterson Amumpaire, Joy Acen, Shamim Katusabe, Joan Tusabe, Joshua Epuitai.

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
