## [Decision Letter · Decision Letter 0]

Dear Dr. Asiimwe,

Thank you for submitting your manuscript to PLOS ONE. After careful consideration, we feel that it has merit but does not fully meet PLOS ONE’s publication criteria as it currently stands. Therefore, we invite you to submit a revised version of the manuscript that addresses the points raised during the review process.

We look forward to receiving your revised manuscript.

Kind regards,

David Chibuike Ikwuka, Ph.D.

Academic Editor

PLOS ONE

“Brenda Nabawanuka obtained funding from Mountains of the Moon University Research & Innovation Fund to conduct this study.”

“We acknowledge the financial support obtained from the Mountains of the Moon University Research and Innovation Fund.

We are grateful for the support provided by Fort Portal Regional Referral Hospital's administration and staff, especially the ones working in the postnatal ward. We also want to express our gratitude to our research assistants who reviewed documents to obtain the data.”

“Brenda Nabawanuka obtained funding from Mountains of the Moon University Research & Innovation Fund to conduct this study.”

Reviewers' comments:

Reviewer's Responses to Questions

**Comments to the Author**

1. Is the manuscript technically sound, and do the data support the conclusions?

Reviewer #1: Yes

Reviewer #2: Yes

2. Has the statistical analysis been performed appropriately and rigorously?

Reviewer #1: Yes

Reviewer #2: Yes

3. Have the authors made all data underlying the findings in their manuscript fully available?

Reviewer #1: Yes

Reviewer #2: Yes

4. Is the manuscript presented in an intelligible fashion and written in standard English?

Reviewer #1: Yes

Reviewer #2: Yes

Reviewer #1: The study is important and highlight the obstetric complications in underprivileged communities especially in Africa. We need more studies from that region about maternal morbidities and mortalities. I am surprised with the high percentage of the maternal sepsis in this study: 24%, can you please include in the manuscript what criteria were used to define sepsis?

Reviewer #2: Dear Editor

I appreciate your efforts in highlighting such an important topic. Its well written and you have managed to fulfill all the basic requirements of the journal. The methodology section is detailed and can easily be replicated. The ethical concerns are addressed. The statistical analysis is rigorously done.

It will contribute to existing literature

**Do you want your identity to be public for this peer review?** For information about this choice, including consent withdrawal, please see our Privacy Policy

Reviewer #1: No

Reviewer #2: **Yes: ** SAIDA ABRAR

---

## [Author Response · Author response to Decision Letter 1]

31 Jan 2025

Comment from reviewer #1: The study is important and highlight the obstetric complications in underprivileged communities especially in Africa. We need more studies from that region about maternal morbidities and mortalities. I am surprised with the high percentage of the maternal sepsis in this study: 24%, can you please include in the manuscript what criteria were used to define sepsis?

Response: Thank you so much for this observation, Puerperal sepsis was considered in this study as a mother with a laboratory-based diagnosis with positive bacterial culture results during the course of postnatal admission. Please see see lines 130 to 133.

Reviewer #2: I appreciate your efforts in highlighting such an important topic. Its well written and you have managed to fulfill all the basic requirements of the journal. The methodology section is detailed and can easily be replicated. The ethical concerns are addressed. The statistical analysis is rigorously done.

It will contribute to existing literature

Response: Thank you so much for the appreciation.

---

## [Editor Report · Decision Letter 1]

Dear Dr. Asiimwe,

Thank you for submitting your manuscript to PLOS ONE. After careful consideration, we feel that it has merit but does not fully meet PLOS ONE’s publication criteria as it currently stands. Therefore, we invite you to submit a revised version of the manuscript that addresses the points raised during the review process.

We look forward to receiving your revised manuscript.

Kind regards,

Sheikh Irfan Ahmed

Academic Editor

PLOS ONE

Journal Requirements:

Additional Editor Comments:

Lines 116-117: All the postnatal hospital records for the previous six months prior data collection were identified and we selected all files that fulfilled the inclusion and exclusion criteria

Comment: The methods section in abstract mentioned that patient files of 180 postnatal mothers who were admitted at Fort Portal Regional Referral Hospital from 20 February, 2024 to 01 April, 2024 were included in the study? Please explain inclusion and exclusion criteria in study population section

129-131: Puerperal sepsis was considered in this study as a mother with a laboratory-based diagnosis with positive bacterial culture results during the course of postnatal admission

Comment: Any abnormal vital signs threshold, SIRS, qSOFA or SSC criteria were considered for diagnosis?

Line: 134 Data collection was conducted between 20 February 2024 and 01 April, 2024.

Comment: Is this the data collection period or patient admission period? Please refer to methods section

Clinically

151-152: plausible variables and those with a p value of ≤0.2 at bivariate analysis were taken for multivariate analyses to adjust for confounding.

Comment: plausible variables and those with a p value of ≤0.2 at bivariate analysis were taken for multivariate analyses to adjust for confounding variables.

---

## [Author Response · Author response to Decision Letter 2]

20 Jun 2025

Comment: The methods section in abstract mentioned that patient files of 180 postnatal mothers who were admitted at Fort Portal Regional Referral Hospital from 20 February, 2024 to 01 April, 2024 were included in the study? Please explain inclusion and exclusion criteria in study population section.

Response: Thank you so much for this comment, the inclusion and the exclusion criteria has been included in the population section. Please see lines 111 to 112.

Comment: Any abnormal vital signs threshold, SIRS, qSOFA or SSC criteria were considered for diagnosis?

Response: Thank you so much for this concern, however in our study we only used the positive bacterial culture results to define puerperal sepsis.

Comment: Is this the data collection period or patient admission period? Please refer to methods section

Response: Thank you for this response, however the period of 20th of February 2024 to 01st of April 2024 was the data collection period. The admission was all those mothers who admitted in postnatal 6 months prior to data collection as indicated in the study population section. Please see lines 108 to 112.

---

## [Editor Report · Decision Letter 2]

Prevalence and Factors Associated with Puerperal Sepsis Among Postnatal Women at a Tertiary Referral Hospital in Western Uganda

PONE-D-24-39427R2

Dear Dr. Asiimwe,

We’re pleased to inform you that your manuscript has been judged scientifically suitable for publication and will be formally accepted for publication once it meets all outstanding technical requirements.

Kind regards,

Sheikh Irfan Ahmed

Academic Editor

PLOS ONE
---

## [Editor Report · Acceptance letter]

PONE-D-24-39427R2

PLOS ONE

Dear Dr. Asiimwe,

I'm pleased to inform you that your manuscript has been deemed suitable for publication in PLOS ONE. Congratulations! Your manuscript is now being handed over to our production team.

Kind regards,

on behalf of

Dr. Sheikh Irfan Ahmed

Academic Editor

PLOS ONE